# Adjuvant Lipoic acid Injection in Sepsis treatment in China (ALIS study): protocol for a randomised, single-blind, placebo-controlled trial

Linhui Hu ![ORCID],[1,2] Xinjuan Zhou,[2,3] Jinbo Huang,[2] Yuemei He,[2,3] Quanzhong Wu,[4] Xiangwei Huang,[1] Kunyong Wu,[5] Guangwen Wang,[5] Sinian Li,[6] Xiangyin Chen,[4] Chunbo Chen ![ORCID] [7]

LH, XZ, JH, YH and QW are joint first authors.

For numbered affiliations see end of article.

**Correspondence to**
Chunbo Chen;
gghccm@163.com

## ABSTRACT

**Introduction** Sepsis is a life-threatening immune disorder resulting from an dysregulated host response to infection. Adjuvant therapy is a valuable complement to sepsis treatment. Lipoic acid has shown potential in attenuating sepsis-induced immune dysfunction and organ injury in vivo and in vitro studies. However, clinical evidence of lipoic acid injection in sepsis treatment is lacking. Hence, we devised a randomised controlled trial to evaluate the efficacy and safety of lipoic acid injection in improving the prognosis of sepsis or septic shock patients.

**Methods and analysis** A total of 352 sepsis patients are planned to be recruited from intensive care units (ICUs) at eight tertiary hospitals in China for this trial. Eligible participants will undergo randomisation in a 1:1 ratio, allocating them to either the control group or the experimental group. Both groups received routine care, with the experimental group also receiving lipoic acid injection and the control group receiving placebo. The primary efficacy endpoint is 28-day all-cause mortality. The secondary efficacy endpoints are as follows: ICU and hospital mortality, ICU and hospital stay, new acute kidney injury in ICU, demand and duration of life support, Sequential Organ Failure Assessment (SOFA)/Acute Physiology and Chronic Health Evaluation II (APACHE II) and changes from baseline (ΔSOFA/ΔApache II), arterial blood lactate (LAC) and changes from baseline (ΔLAC), blood procalcitonin, high-sensitivity C-reactive protein, interleukin-2 (IL-2), IL-4, IL-6, IL-10, tumour necrosis factor-α (TNF-α) and interferon-γ (IFN-γ) and changes from baseline on day 1 (D1), D3, D5 and D7. Clinical safety will be assessed through analysis of adverse events.

**Ethics and dissemination** The study was approved by the Ethics Committee of Maoming People's Hospital (approval no. PJ2020MI-019-01). Informed consent will be obtained from the participants or representatives. The findings will be disseminated through academic conferences or journal publications.

**Trial registration** ChiCTR2000039023.

## STRENGTHS AND LIMITATIONS OF THIS STUDY

⇒ The trial utilises a randomised controlled design with a substantial sample size.
⇒ The trial design mitigates the influence of confounding factors, such as terminal diseases.
⇒ Adequate evidence from animal and human research supports the hypothesis that lipoic acid could protect organs against sepsis injury.
⇒ This trial follows a single-blind study design rather than a double-blind design.

## INTRODUCTION

Sepsis, characterised by acute organ dysfunction resulting from a dysregulated host response to infection, is a multifaceted disorder that poses a significant mortality risk among patients in non-coronary intensive care units (ICUs).[1–3] Despite the advancements in sepsis management over the past decade, the mortality rate still exceeds 30%, reaching up to 60% in cases of septic shock.[4] In high-income countries, the population incidence rate of sepsis was 437 cases per 100 000 person-years, resulting in approximately 2.8 million deaths attributed to sepsis annually.[5] Jiang *et al*[6] reported a standardised sepsis incidence and mortality rate of 461 and 79 cases per 100 000 person-years, respectively, in China. The incidence is expected to rise further due to the ageing Chinese population, leading to an increased financial burden associated with treatment. Consequently, prioritising the management of sepsis and septic shock becomes crucial for ensuring patient safety.

The primary treatment approach for sepsis emphasises timely intervention, which includes removing the source of infection, administering appropriate antibiotics and restoring tissue perfusion through fluid resuscitation and vasoactive drugs.[7–9] Indeed, sepsis is a syndrome rather than a specific disease, which highlights the importance of implementing adjuvant therapies that target

its underlying pathophysiological mechanisms beyond primary treatment. The 2021 guidelines only provide weak recommendations for adjuvant therapies in sepsis management,[10] such as the use of intravenous corticosteroids. Consequently, there is a pressing need for safe, effective and affordable adjuvant treatments with minimal adverse effects.

The exact mechanisms of sepsis-induced cell injury and organ dysfunction are not fully understood. However, it is widely recognised that simultaneous imbalanced activation of proinflammatory response (leading to cellular and tissue damage) and anti-inflammatory response (leading to immune system impairment) are associated with sepsis.[11–13] Moreover, sepsis progression has been linked to the imbalance between reactive oxygen species (ROS) and antioxidant enzymes. One compound that has shown unique antioxidant properties is lipoic acid (LA), also known as 1,2-dithiolane-3-pentanoic acid. LA is a naturally occurring dithiol compound that is enzymatically synthesised in the mitochondria from octanoic acid. LA chelates several metal ions[14] and scavenges ROS and reactive nitrogen species (RNS) both in vitro and in vivo.[15] These actions enable LA to regenerate antioxidant molecules like glutathione, vitamin C and vitamin E.[16] Remarkably, due to its small chemical structure, LA exhibits solubility in both water and lipids. This unique characteristic enables LA to efficiently access and permeate throughout the entire body, facilitating its favourable effects.

In a study conducted by Zhang et al,[17] it was demonstrated that LA effectively attenuates lipopolysaccharide (LPS)-induced acute inflammatory responses both in vitro and in vivo. This attenuation is achieved through the activation of the phosphoinositide 3-kinase (PI3K)/protein kinase B (Akt) signalling pathway. Multiple studies[18–21] have demonstrated that LA possesses anti-inflammatory properties and can effectively reduce LPS-induced inflammation. LA achieves this by inhibiting the activation of the nuclear factor κB (NF-κB) signalling pathway and suppressing the excessive production of proinflammatory factors. These actions contribute to the attenuation of sepsis-induced inflammation and oxidative damage in organs, highlighting the potential of LA as a therapeutic intervention for sepsis. Furthermore, LA has shown significant benefits in improving diabetic neurovascular and metabolic abnormalities.[22] Additionally, it has been implicated in cardiovascular protection and exerting anti-inflammatory effects.[23–26] This suggests that LA may have broad therapeutic applications beyond sepsis treatment. Considering its anti-oxidative properties, LA holds promise as a potential adjuvant for sepsis treatment. Its ability to protect tissues from oxidative stress and prevent sepsis-induced organ dysfunction makes it a candidate worth exploring further.

Indeed, LA has been investigated in various therapeutic contexts, including weight management,[27 28] multiple sclerosis,[29] male fertility[30] and type 2 diabetes,[31 32] showing promising results. However, when it comes to sepsis-induced organ dysfunction, most studies have primarily focused on animal models, particularly rats.[17 33 34] As of today, there is a lack of human studies exploring the therapeutic effects of LA in sepsis treatment.

To address this gap in knowledge, we have designed a multicentre, single-blinded, randomised controlled trial (RCT) in China. The objective of this trial is to assess the efficacy and safety of LA injection compared with a placebo in patients diagnosed with sepsis or septic shock in ICUs. We aim to investigate the preventive and therapeutic effects of LA on suppressing inflammation, attenuating organ dysfunction, shortening ICU or hospital stay and improving patient outcomes, including mortality. By conducting this RCT, we hope to provide valuable insights into the potential benefits of LA in sepsis treatment.

## METHODS AND ANALYSIS
### Study design and setting
The ALIS ('Adjuvant Lipoic acid Injection in Sepsis treatment in China') study is a multicentre, single-blind, randomised, placebo-controlled clinical trial conducted in tertiary hospitals in China. The protocol and informed consent forms are required to undergo a thorough review and approval by the applicable ethical committees (ECs) to ensure compliance with scientific standards and human subject protection regulations at each participating centre. The study involves various departments, including general ICU, neurological ICU, surgical ICU and emergency ICU. Detailed information about the intended participating centres can be found in table 1. The enrolment period for this study is estimated to be 2 years, and the trial will conclude with the last follow-up. The study has been registered,[35] ensuring transparency and accessibility of the study information. The study design and protocol adhere to the Standard Protocol Items: Recommendations for Interventional Trials reporting guidelines.[36]

### Protocol amendments
Protocol amendments will be recorded, including a concise description of the modification and a reference (date and number), in cases where changes to the existing protocol have a significant impact on subject safety, study scope or scientific quality. The coordinating investigator has a responsibility to inform all investigators, the reviewing EC and other relevant parties about these protocol amendments as required.

### Eligibility criteria
All trial centres will adhere to the same participant inclusion criteria. The inclusion criteria are as follow:
1. Meeting the diagnostic criteria for sepsis and septic shock, as established by the American Critical Care Medical Association/European Critical Care Medical Association (Sepsis-3 criteria).[37]
2. Age between 18 and 80 years.

**Table 1** Research settings and names of each ethics committee

| Research setting | Ethics committee name | Hospital rank and location |
|---|---|---|
| Maoming People's Hospital | Medical Ethics Committee of Maoming People's Hospital | Tertiary, Maoming, Guangdong, China |
| Guangdong Provincial People's Hospital | Medical Ethics Committee of Guangdong Provincial People's Hospital | Tertiary, Guangzhou, Guangdong, China |
| Affiliated Hospital of Guangdong Medical university | Medical Ethics Committee of Affiliated Hospital of Guangdong medical university | Tertiary, Zhanjiang, Guangdong, China |
| Central People's Hospital of Zhanjiang | Medical Ethics Committee of Central People's Hospital of Zhanjiang | Tertiary, Zhanjiang, Guangdong, China |
| Huizhou Central Hospital | Medical Ethics Committee of Huizhou Central Hospital | Tertiary, Huizhou, Guangdong, China |
| Xiaolan People's Hospital of Zhongshan City | Medical Ethics Committee of Xiaolan People's Hospital of Zhongshan | Tertiary, Zhongshan, Guangdong, China |
| Third People's Hospital of Huizhou | Medical Ethics Committee of the Third People's Hospital of Huizhou | Tertiary, Huizhou, Guangdong, China |
| Yunfu People's Hospital | Medical Ethics Committee of Yunfu People's Hospital | Tertiary, Yunfu, Guangdong, China |

3. Within 48 hours of receiving a diagnosis of sepsis or septic shock.
4. Having provided signed informed consent.

The exclusion criteria for participant selection are as follows:

1. Female patients who are pregnant or lactating.
2. Presence of persistent infection sources that cannot be effectively eliminated through procedures such as puncture, drainage, debridement, or other surgical operations.
3. Participation in another clinical trial.
4. Current treatment with other medications that have antioxidant effects, such as vitamin C and vitamin E.
5. Accompanied by non-infectious but life-threatening conditions, such as advanced tumour, severe craniocerebral injury, uncontrollable massive bleeding, cardiogenic shock, etc.
6. Lack of approval for comprehensive and active life support treatment.
7. Allergy to LA or similar active agents (eg, vitamin B), or previous intolerance to the recommended dose of LA.

### Interventions

The participants will be randomly assigned to either the placebo group or the LA group and treated according to their assigned treatment plan. In the placebo group, patients will receive a placebo consisting of 250 mL of normal saline. In the treatment group, patients will receive a solution containing an LA injection of 600 mg (provided by Yabao Pharmaceutical Group Co.) dissolved in 250 mL of normal saline. To prevent exposure to light, both the LA and the placebo solutions will be wrapped in aluminium platinum paper. It is important that the intravenous drip time does not fall short of 30 min. Concomitant care include standard sepsis bundle treatment and other routine care. The critical care physicians are encouraged to adhere to the updated Surviving Sepsis Campaign guidelines,[7] although the final decision is at the discretion of the physicians as per institutional guidelines. The number and percentage of patients in both the treatment and placebo groups who deviated from the sepsis bundle treatment will be documented.

### Evaluation and outcomes

Participants in the study will undergo clinical evaluation and laboratory testing, as outlined in online supplemental table 1. The collected data will include demographic information such as age and gender, disease diagnosis and progression, interventions and prognosis, as well as adverse events (AEs). The disease diagnosis data will encompass primary and concomitant diseases, blood culture results, white blood cell count and liver and kidney biochemistry. Disease progression data will involve baseline severity scores and serum biomarkers, including Sequential Organ Failure Assessment (SOFA) score, Acute Physiology and Chronic Health Evaluation II (APACHE II) score, procalcitonin (PCT), high-sensitivity C-reactive protein (hs-CRP), interleukin 2 (IL-2), IL-4, IL-6, IL-10, tumour necrosis factor-$\alpha$ (TNF-$\alpha$) and interferon-$\gamma$ (IFN-$\gamma$). Follow-up values on specific days (day 1 (D1), D3, D5, D7) after randomisation will also be recorded, along with the changes from baseline.

Data related to disease intervention will include the requirement and duration of life support measures like mechanical ventilation, vasoactive drugs and continuous renal replacement therapy. Disease prognosis data will encompass 28-day all-cause mortality, ICU mortality, in-hospital mortality, incidence of new-onset acute kidney injury (AKI) after ICU admission, ICU stay and length of hospital stay. Throughout the study, data on organ dysfunction, occurrence of AKI and safety events will be continuously monitored.

The primary efficacy endpoint of the study is 28-day all-cause mortality. The secondary efficacy endpoints are as follows:

1. ICU mortality
2. Hospital mortality
3. Duration of ICU and hospital stay
4. Incidence of new AKI in ICU
5. SOFA/APACHE II scores on specific days (D1, D3, D5 and D7) and their changes from baseline (ΔSOFA/ΔApache II)
6. Demand and duration of life support measures (vasopressors, invasive mechanical ventilation and continuous renal replacement therapy).
7. Lactate concentration of arterial blood on specific days (D1, D3, D5 and D7) and their changes from baseline (ΔLAC).
8. Levels of PCT, hs-CRP, IL-2, IL-4, IL-6, IL-10, TNF-α and IFN-γ on specific days (D1, D3, D5, and D7) and their changes from baseline.

   The detection of levels of IL-2, IL-4, IL-6, IL-10, TNF-α, and IFN-γ is not mandatory if unavailable.

The safety endpoint of the study includes the assessment of any AEs, including serious adverse events (SAEs) and adverse drug reactions (ADRs). These ADRs may manifest as various symptoms, such as head swelling, dyspnoea, convulsion, diplopia, purpura, bleeding tendency, allergies, and others.

During clinical research, it is essential for researchers to closely observe any AEs experienced by subjects. When AEs are identified, researchers should promptly take appropriate measures based on the subjects' conditions. Regardless of whether the AEs are associated with LA injection or not, it is necessary for researchers to diligently record them in detail within the case report form (CRF). The recorded information regarding AEs should encompass a description of the events, their occurrence time, severity, duration, measures taken, final results and outcomes. These records serve the purpose of analysing the potential correlation between AEs and LA injection, and they need to be signed and dated. In the event of encountering any SAEs, it is mandatory to report them to the principal investigator of the trial, the ethics committee (EC) and the state supervision agency within 24 hours of their occurrence. When an AE is identified, the researcher can administer necessary treatment measures considering the subject's condition, such as dose adjustment or temporary suspension of research drug usage. The decision regarding whether to terminate the study will depend on the circumstances. In case of an SAE occurrence, immediate action must be taken by the researcher to safeguard the safety of the subjects.

### Participant timeline

The follow-up period for this trial is 28 days. At the final visit of the last patient, data regarding the clinical status of all participants who have either completed the trial or withdrawn will be collected.

### Sample size

In this study, the primary efficacy endpoint was the 28-day all-cause mortality rate. A large-scale epidemiological survey conducted in the ICU revealed that the control group had a 38.2% 28-day all-cause mortality rate. In an RCT comparing the LA injection treatment group, the relative risk reduction was approximately 0.33.[38] The anticipated 28-day all-cause mortality rate for the treatment group was expected to be 22.9% with a one-sided test, using α=0.025 and β=0.2 (power=80%). The threshold value, Δ, was set at 0.

Participants are randomised at a ratio of 1:1, resulting in a calculated sample size of 160 for each group using a superiority test (where a lower rate is considered superior). Taking into account the anticipated dropout rate of around 10%, the sample size was increased to 176 cases in each group.

### Recruitment

To achieve adequate participant enrolment and reach the target sample size, the trial is planned to be conducted in multiple tertiary hospitals as outlined in table 1. The enrolment period is estimated to span 2 years. The study successfully recruited its first participant on 26 August 2021. A well-trained study coordinator has been appointed at each participating centre to meticulously screen all potentially eligible patients based on the predefined eligibility criteria. The anticipated timeline for completion of the study is set for 1 September 2024.

### Consent and assent

The clinical investigator has the responsibility to obtain informed consent from every research subject prior to their participation in the study. Detailed information about the informed consent can be found in online supplemental appendix 1. In cases where a research subject is unable to provide consent due to a critical illness or disability, informed consent is obtained from authorised surrogates. However, it is important to note that obtaining prior approval from the EC is necessary for this process. Our intention is solely to obtain consent for the use of data and samples related to the research question outlined in this protocol. We do not have any plans to utilise participant data or biological samples for ancillary studies.

### Allocation

In this study, participants who have been diagnosed with sepsis based on the Sepsis-3 criteria are included in the screening process. Those who meet the eligibility criteria will be randomly assigned to either the treatment or control group using a 1:1 allocation. The randomisation will be conducted using a computer-generated block randomisation schedule with a block size of 8, stratified by site. Each block will consist of four patients receiving the LA injection and four patients receiving a placebo. The randomisation method and block size will not be blinded until all data analysis is completed. The investigators

involved in the study will not participate in data collection. The randomisation centre, through net verification, will assign the screened patients at each hospital in a random manner.

To ensure the overall quality and integrity of the clinical trial, code breaks should only occur in exceptional circumstances when knowledge of the actual treatment is absolutely necessary for further patient management. Investigators are encouraged to consult with medical advisors if they believe that unblinding is necessary. Any instances of code breaks must be reported, along with the reason, on the corresponding CRF page. It is important to note that unblinding should not automatically result in the discontinuation of the study drug.

## Data collection and management
To ensure the competence in managing the project, data collectors, researchers and coordinators involved in the study will receive a thorough training in advance. The coordinators at each centre will utilise an electronic data acquisition system for inputting the CRF information. The specific system being used has obtained copyright for its computer software on 6 April 2022 (software no. 9387742). After data entry, a designated individual will regularly perform accuracy checks to ensure the integrity of the entered data. If any data revisions are necessary, the modified data will undergo review by the data manager. Data access will be strictly restricted to authorised personnel only. The principal investigators at each centre will have the ability to log into the system and view the data, but access to full information will only be granted with permission from the coordinating investigators of the research. It is emphasised that all research-related documents will be treated as highly confidential and stored securely at the study site.

## Data monitoring
An independent Data and Safety Monitoring Board () will assume responsibility for data monitoring and blinded analysis in this trial. DSMB committee members will regularly review the research data to ensure its integrity. Furthermore, the trial will undergo examination by the medical EC of each participating centre, which will assess its adherence to ethical principles and guidelines.

## Analyses sets
The analysis of the trial will be conducted following the 'intention-to-treat' (ITT) principle. This means that all patients who have undergone randomisation, regardless of whether they completed the trial or received the designated treatment, will be included in the analysis according to their original assigned groups. The randomisation information will be retained to the maximum extent possible, ensuring that the integrity of the randomisation process is maintained during the analysis.

### Intention-to-treat set (ITTS)
The principle of ITT analysis is adopted to retain the randomised information effectively and attribute differences in trial results to treatment variations. This ensures that the assessment of treatment effects, specifically in the case of LA treatment, is conducted in the most accurate manner possible. The ITT set (ITTS) consists of all patients who were randomised, regardless of whether they received their designated treatment or completed the full trial. All patients in the ITTS remain in their original assigned groups for analysis. By adhering to the ITT principles, the analysis maintains the integrity of the randomisation process and provides a comprehensive evaluation of the treatment's effectiveness, accounting for the real-world scenario.

### Per protocol set (PPS)
The PPS includes all participants who meet the eligibility criteria and receive treatment as per their assigned randomisation. These participants demonstrate good compliance with the trial protocol, including being prepared to receive treatment, undergoing measurements for primary efficacy endpoints and ensuring that the necessary information is filled in the CRF. The PPS is specifically utilised to analyse the primary efficacy endpoints of the study. By focusing on participants who adhere closely to the trial protocol, the PPS analysis provides insights into how the treatment performs under optimal conditions and can help evaluate the treatment's efficacy in a controlled setting.

### Security set (SS)
The SS comprises all actual cases that have received at least one administration of the assigned fluid as per randomisation. This set includes participants for whom safety endpoints are recorded, allowing for the assessment of AEs. The incidence of AEs is calculated based on the number of patients included in the safety set. By analysing the safety data from this set, researchers can evaluate the occurrence and severity of AEs related to the administered fluid, providing important insights into the safety profile of the treatment.

## Statistical analysis
All analyses, including the primary analyses, will be conducted in accordance with the ITT principle. The data will be analysed using R software V.3.3.3 (R Foundation for Statistical Computing, Vienna, Austria) and RStudio V.1.0.136 (RStudio, Boston, MA, USA). Descriptive statistics will be used to describe the data's distribution, with continuous data presented as mean (SD) or median and IQR, depending on the data characteristics. Categorical data will be reported as number and percentage. Group comparisons will be conducted using group t-tests or analysis of variance for continuous data assuming normality, and non-parametric tests will be used if normality assumptions are not met. Categorical results will be compared using $\chi^2$ test or Fisher's exact test. The statistical significance level for all tests will be set at a p value of less than 0.05.

## Interim analysis
No interim analysis is scheduled for this trial.

## Ancillary and post-trial care
In each participating study centre, all patients will receive treatment, monitoring and routine assessments related to the recovery process in accordance with the Surviving Sepsis Campaign guidelines.

## Patient and public involvement
None.

## Ethics and dissemination
### Ethics approval
This study has received approval from the Ethics Committee of Maoming People's Hospital, China. The approval (no. PJ2020MI-019-01) was granted on 25 September 2020. Prior to enrolment in the study, all participants will be required to provide informed consent that has been approved by the ethics committee. Once enrolled, the study will be conducted in strict adherence to the clinical research plan and in accordance with the principles outlined in the Declaration of Helsinki.

### Informed consent forms and privacy protection
In this study, individuals will not be eligible to participate in the study until they have obtained a signed and written informed consent form. Throughout the participants' involvement in the study, any updated versions of the informed consent forms and written information will be provided to them. The informed consent forms are considered important documents of the clinical research and will be retained for future reference. If any changes are made to the informed consent form during the study, the revised version can only be used after obtaining written approval from the ethics committee, and participants will be required to resign the revised version.

Participants in this project acknowledge and endorse the guidelines established to safeguard the privacy of subjects from any infringement. Throughout the entire duration of the study, the original data of the participants will only be linked to the clinical research database or files of the participating units using a unique identification number. Limited subject characteristics, such as gender, age, date of birth and initials of the subject's name, may be used to verify the accuracy of the subject's information and their unique identification number, with the approval of applicable laws and regulations. These measures ensure the confidentiality and protection of the participants' data.

### Dissemination
The results of this study will be shared through presentations at relevant national and international conferences. Additionally, the findings will be submitted for publication in peer-reviewed journals. This dissemination approach ensures that the research outcomes reach a wide audience and undergo rigorous evaluation by experts in the field.

## DISCUSSION
Sepsis is a severe medical condition characterised by dysfunction of organs caused by the body's uncontrolled response to an infection. It is defined by an acute increase in the SOFA score of 2 points or more.[39] Septic shock, on the other hand, is a subset of sepsis. It refers to a condition where there are profound circulatory, cellular and metabolic abnormalities that result in hypotension, requiring the administration of vasopressor medications to maintain a mean arterial blood pressure above 65 mm Hg. In septic shock, patients also exhibit elevated levels of serum lactate concentration, exceeding 2 mmol/L, despite adequate fluid resuscitation efforts.

In 2017, the World Health Assembly recognised sepsis and septic shock as global health priorities due to their high prevalence and alarming mortality rates.[40] Moreover, many sepsis survivors suffer from long-term complications, underscoring the urgent need to address this issue. While most data on sepsis come from high-income countries, where 2.8 million deaths are attributed to sepsis annually, a study conducted by Du *et al*[6] reported standardised incidence and mortality rates of 461 and 79 cases per 100 000 person-years, respectively, in China. However, it is plausible that these numbers might be higher due to lower awareness of sepsis and limited access to intensive care and organ support. Despite being a global priority, the available research and evidence on sepsis disproportionately come from high-income countries. Therefore, there is a critical need for more research, particularly in low-income and middle-income countries like China, which has a population exceeding 1.4 billion. This research should focus on identifying pharmacological interventions that can improve outcomes by targeting the underlying pathophysiological mechanisms of the disease.

Sepsis is characterised by a dysregulated systemic host response to infection. One well-established pathway of immune activation in response to infection involves the recognition of highly conserved microbial pathogen-associated molecular patterns by pattern-recognition receptors, including toll-like receptors, present on innate immune system cells. This recognition leads to the activation of NF-κB and neutrophils, triggering the release of proinflammatory and anti-inflammatory mediators. Cytokines such as TNF-α, IL-1, IL-2, IL-6 and IL-8, among others, promote neutrophil–endothelial cell adhesion and activate the complement and clotting cascades, contributing to what is commonly referred to as the 'inflammatory storm'.

The PI3K/Akt pathway is a well-conserved family of signal transduction enzymes that plays a crucial role in regulating cellular proliferation and survival. However, there are notable variations in the anti-inflammatory function of the PI3K/Akt pathway across different cell types. Depending on the specific stimulus and cell type involved, it can act as either a positive[41–43] or negative regulator[44–46] of NF-κB activation and cytokine production. Indeed, an increasing body of evidence indicates that the PI3K/Akt pathway serves as a critical negative feedback regulator

of exaggerated innate immune and toll-like receptor-mediated proinflammatory responses.[44–51] Multiple studies[44 47–49 52 53] support the role of the PI3K/Akt pathway in limiting inflammation in endotoxemia and sepsis. Inhibition of PI3K/Akt signalling has been found to enhance LPS-induced activation of NF-κB and AP-1, along with increased gene expression of TNF-α and tissue factor in cultured cells.[44] NF-κB has been extensively documented to play a significant role in the transcriptional regulation of inflammatory genes induced by LPS, which contribute to the progression of septic shock, multiple organ failure, and mortality.[54–57] Clinically, increased NF-κB binding activity has been observed in patients with acute inflammation and sepsis, demonstrating a correlation with clinical severity and mortality.[58 59] As a result, LA could potentially suppress septic inflammation by modulating the PI3K/Akt pathway.

Oxidative stress, a significant promoter and mediator of the systemic inflammatory response, is considered crucial in the pathophysiology of sepsis.[60] The early phase of sepsis is characterised by an excessive formation of ROS and RNS, including superoxide and nitric oxide. Excessive production of these radical species during inflammation is associated with sepsis development, leading to marked oxidative stress, cellular injury, disruption of antioxidant systems and mitochondrial dysfunction, ultimately resulting in organ damage.[61 62] In normal conditions, a complex network of antioxidant defences can counteract oxidative stress and protect mitochondria. However, severe oxidative stress occurs in sepsis, overwhelming the host's endogenous antioxidant defences and disrupting cellular redox balance. It has been proposed that targeted antioxidant therapy aimed at mitochondria, which enhances endogenous antioxidant defences, holds potential for protecting against radical-induced injury.[63 64] Thus, LA may be beneficial in preventing organ injury by acting as a suppressor of ROS.

In summary, LA, as an agent that modulates inflammatory cascades through the aforementioned mechanisms, shows potential effectiveness in balancing the immune response and reducing organ impairment in septic patients. Therefore, we have developed a study protocol for an RCT to investigate the efficacy and safety of LA injection as a treatment for sepsis in China.

This study design possesses several strengths. First, the utilisation of a large sample size in this RCT can enhance our understanding of the efficacy of LA, a commonly used adjuvant drug in sepsis treatment lacking evidence-based support. Second, the exclusion of terminal disease as a confounding factor is addressed in this study. Sepsis can occur in patients already succumbing to other causes (eg, terminal cancer), and inherent mortality associated with sepsis is a characteristic of the syndrome itself. Third, the hypothesis that LA provides organ protection against sepsis injury is backed by substantial evidence from both animal and human research. Nonetheless, there are certain limitations that should be acknowledged. First, the relatively limited number of participating centres may

introduce selection bias, potentially affecting the generalisability of the research findings to other contexts or settings. Second, this study design adopts a single-blind approach, which could introduce diagnostic bias by the investigators, potentially resulting in treatment imbalances between the control and treatment groups.

In conclusion, the ALIS study is a large-sample-sized, multicentre, single-blinded, randomised, parallel-group, placebo-controlled trial in China. It is designed based on robust evidence from basic and clinical studies. The anticipated outcomes of this trial have the potential to offer evidence-based recommendations for sepsis prevention in critically ill patients.

**Author affiliations**
[1]Department of Critical Care Medicine, Maoming People's Hospital, Maoming, Guangdong, China
[2]Clinical Research Center, Center of Scientific Research, Maoming People's Hospital, Maoming, Guangdong, China
[3]Department of Cardiology, Maoming People's Hospital, Maoming, Guangdong, China
[4]Department of Surgery Intensive Care Medicine, Maoming People's Hospital, Maoming, Guangdong, China
[5]Center of Scientific Research, Maoming People's Hospital, Maoming, Guangdong, China
[6]Department of Neurocritical Care Unit, Maoming People's Hospital, Maoming, Guangdong, China
[7]Department of Critical Care Medicine, Shenzhen People's Hospital, The Second Clinical Medical College of Jinan University, The First Affiliated Hospital of Southern University of Science and Technology, Shenzhen, 518020, China

**Contributors** CC conceived of the study. LH, XZ, JH, YH and QW initiated the study design and XH, KW, GW, SL and XC helped with implementation. CC and LH are grant holders. LH, YH and XZ provided statistical expertise in clinical trial design. LH and XZ drafted the manuscript. CC helped draft and revise the manuscript. All authors contributed to refining the study protocol and approved the final manuscript.

**Funding** Author CC is currently receiving a grant (#MaoRenCaiBan [2020]24) from the Office of Talent Work Leading Team in Maoming, a grant (#201803010058) from the Guangzhou Science and Technology Program, a grant (#82172162) from the National Natural Science Foundation of China and a grant (#DFJH2020028) under the Major Program of Summit Project, Guangdong Province High-level Hospital Construction Project of Guangdong Provincial People's Hospital, Guangdong Academy of Medical Sciences. Author LH is currently receiving a grant (#2020YJ01) from the Emergent Science and Technology Project for Prevention and Treatment of Novel Coronavirus Pneumonia, a grant (#zx2020017) from the High-level Hospital Construction Research Project of Maoming People's Hospital, and a grant (#SY2021005) from Excellent Young Talents Project of Maoming People's Hospital, a grant (#B2022246) from Medical Research Fund of Guangdong Province and a grant (#2021KJZXZJYX003) from Special Science and Technology Fund of Maoming City. The study was supported by the Special Science and Technology Fund of Maoming City. The funders had no role in the design and conduct of the study; collection, management, analysis and interpretation of the data; preparation, writing, review or approval of the report and decision to submit the report for publication.

**Competing interests** None declared.

**Patient and public involvement** Patients and/or the public were not involved in the design, or conduct, or reporting or dissemination plans of this research.

**Patient consent for publication** Not required.

**Provenance and peer review** Not commissioned; externally peer reviewed.

**ORCID iDs**
Linhui Hu http://orcid.org/0000-0002-1649-6624
Chunbo Chen http://orcid.org/0000-0001-5662-497X

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
