## [Reviewer comments · BMJ Open]

ARTICLE DETAILS

TITLE (PROVISIONAL)	Adjuvant Lipoic acid Injection in Sepsis treatment in China (ALIS study): protocol for a randomized, single-blind, placebo-controlled trial
AUTHORS	Hu, Linhui; Zhou, Xinjuan; Huang, Jinbo; He, Yuemei; Wu, Quanzhong; Huang, Xiangwei; Wu, Kunyong; Wang, Guangwen; Li, Sinian; Chen, Xiangyin; Chen, Chunbo

VERSION 1 – REVIEW

REVIEWER	Jun Lyu Jinan University First Affiliated Hospital
REVIEW RETURNED	01-Jun-2023

GENERAL COMMENTS	I am happy for the opportunity to review the manuscript entitled "Adjuvant Lipoic acid Injection in Sepsis treatment in China (ALIS study): study protocol for a randomized controlled trial." The authors have designed a randomized controlled trial to explore the efficacy and safety of clinically used lipoic acid injection in improving the prognosis of patients with sepsis or septic shock. The manuscript presents a clear and concise study protocol, which is well-organized and easy to follow. The study design is appropriate, and the use of a placebo control is commendable. The sample size is adequate, and the inclusion and exclusion criteria are clearly defined. The primary and secondary endpoints are relevant and clinically meaningful. Additionally, the study's hypothesis has adequate scientific evidence from animal and human research, making it a promising prospect. I would like to add that the manuscript appears to adhere well to the SPIRIT guidelines. The protocol has provided comprehensive details and adequately addressed the 33-item checklist of SPIRIT. It is evident that the authors have made a significant effort to develop a robust and transparent study design. The manuscript includes essential information, such as the background, study objectives, eligibility criteria, randomization processes, intervention, outcomes, sample size, statistical analysis plan, and adverse events monitoring. The primary and secondary endpoints are clinically relevant and meaningful, and the study design incorporates a control group, which increases the rigor of the study. Moreover, the manuscript addresses the predefined statistical analysis plan and the handling of missing data, which is consistent with the SPIRIT guidelines. This approach ensures the transparency and validity of the study results.
---

	In conclusion, the manuscript adheres well to the SPIRIT guidelines, ensuring comprehensiveness, transparency, and quality of the study protocol. Following SPIRIT guidelines can help ensure adherence to good clinical practice and increase confidence in the reliability, validity, and generalizability of the study results. The manuscript is well-written and meets the standard of publication in BMJ Open. However, there are a few points that need attention. Firstly, given the high mortality and morbidity of sepsis, it would be essential to know the baseline characteristics of the enrolled patients, such as the severity of illness, comorbidities, and demographic information. These factors may influence the treatment outcomes. Secondly, the authors could consider providing more detailed information about the lipoic acid injection, such as the dosage, administration frequency, and duration. Thirdly, it would be beneficial to discuss the potential side effects of lipoic acid injection and how they will be monitored and reported.
--	--

REVIEWER	Jose Iglesias Hackensack Meridian Jersey Shore University Medical Center, Internal Medicine/Nephrology
REVIEW RETURNED	12-Jun-2023

GENERAL COMMENTS	In my opinion although treatment of patients should be left to individual intensivist. The patients enrolled in study should at least be treated with sepsis bundle if not this should be listed in the limitations. In addition perhaps in the appendix tx and placebo pts that deviated from bundle should be shown for example there were 10% in the treatment group vs 8% in the placebo group that did not receive sepsis bundle.
--

VERSION 1 – AUTHOR RESPONSE

Reviewer 1:

1. Comment:

I am happy for the opportunity to review the manuscript entitled "Adjuvant Lipoic acid Injection in Sepsis treatment in China (ALIS study): study protocol for a randomized controlled trial." The authors have designed a randomized controlled trial to explore the efficacy and safety of clinically used lipoic acid injection in improving the prognosis of patients with sepsis or septic shock.

The manuscript presents a clear and concise study protocol, which is well-organized and easy to follow. The study design is appropriate, and the use of a placebo control is commendable. The sample size is adequate, and the inclusion and exclusion criteria are clearly defined. The primary and secondary endpoints are relevant and clinically meaningful. Additionally, the study's hypothesis has adequate scientific evidence from animal and human research, making it a promising prospect.

I would like to add that the manuscript appears to adhere well to the SPIRIT guidelines. The protocol has provided comprehensive details and adequately addressed the 33-item checklist of SPIRIT. It is evident that the authors have made a significant effort to develop a robust and transparent study design.

The manuscript includes essential information, such as the background, study objectives, eligibility criteria, randomization processes, intervention, outcomes, sample size, statistical analysis plan, and adverse events monitoring. The primary and secondary endpoints are clinically relevant and

meaningful, and the study design incorporates a control group, which increases the rigor of the study. Moreover, the manuscript addresses the predefined statistical analysis plan and the handling of missing data, which is consistent with the SPIRIT guidelines. This approach ensures the transparency and validity of the study results.

In conclusion, the manuscript adheres well to the SPIRIT guidelines, ensuring comprehensiveness, transparency, and quality of the study protocol. Following SPIRIT guidelines can help ensure adherence to good clinical practice and increase confidence in the reliability, validity, and generalizability of the study results.

The manuscript is well-written and meets the standard of publication in BMJ Open. However, there are a few points that need attention. Firstly, given the high mortality and morbidity of sepsis, it would be essential to know the baseline characteristics of the enrolled patients, such as the severity of illness, comorbidities, and demographic information. These factors may influence the treatment outcomes. Secondly, the authors could consider providing more detailed information about the lipoic acid injection, such as the dosage, administration frequency, and duration. Thirdly, it would be beneficial to discuss the potential side effects of lipoic acid injection and how they will be monitored and reported.

Response:

Thank you for your positive feedback on the overall quality of our manuscript. We appreciate your valuable suggestions for further improvement.

Regarding your points, we agree that providing baseline characteristics of the enrolled patients is crucial to better understanding the potential impact of sepsis severity, comorbidities, and demographic factors on treatment outcomes. Indeed, we planned and stated in the Evaluation and Outcomes section that the baseline characteristics and sepsis severity (APACHE II and SOFA score) of the enrolled patients were routinely documented in our design.

Furthermore, we acknowledge the importance of providing detailed information about the lipoic acid injection. In the manuscript, we have indeed included the dosage, administration frequency, and duration of the lipoic acid treatment. This additional information will enhance the clarity and reproducibility of our study.

Lastly, as recommended, we have extracted the previous text on adverse events into a subsection named Adverse Events to clarify the potential side effects of lipoic acid injection and outline our monitoring and reporting procedures for these side effects. This will provide a more comprehensive understanding of the safety profile of the treatment .

We are grateful for your insightful comments, which have significantly strengthened the manuscript.

Reviewer 2:

1. Comment:

In my opinion although treatment of patients should be left to individual intensivist. The patients enrolled in study should at least be treated with sepsis bundle if not this should be listed in the limitations. In addition perhaps in the appendix tx and placebo pts that deviated from bundle should be shown for example there were 10% in the treatment group vs 8% in the placebo group that did not receive sepsis bundle.

Response:

We appreciated your insightful and valuable feedback and suggestions regarding the treatment of patients and potential protocol deviation documentation in our study.

Our intention is to request adherence to the guidelines for the sepsis bundle, based on which individual treatments are also allowed. We apologize for any confusion caused by our previous statement, as it may not have been expressed clearly. We acknowledge the need to improve our clarity in delivering the information to our readers. To address that, in our revised manuscript, we have now clearly stated that all enrolled patients received standard sepsis bundle treatment and other routine care as per institutional clinical practice. This information has been updated in the Interventions subsection of the Methods section to provide transparency and ensure that readers are aware of the treatment provided to the patients (Page 15, Lines 13-16).

We also greatly appreciate your suggestion to present the number and percentage of patients in both the treatment and placebo groups who deviated from the sepsis bundle treatment. Actually, the study team will strictly control the patients who deviated from the sepsis bundle. Those deviated patients will be included in the intention-to-treat set, and excluded from the PP set for analysis. We will report this data in the study flow in future dissemination.

We appreciate your valuable comment, which has helped strengthen the manuscript. We hope these revisions have addressed your concerns and provided additional information for readers to evaluate the treatment outcomes in the context of sepsis bundle adherence. Thank you again for your time and consideration.

VERSION 2 – REVIEW

REVIEWER	Jun Lyu Jinan University First Affiliated Hospital
REVIEW RETURNED	16-Jul-2023
GENERAL COMMENTS	It is OK.